# Supernovae and the Arrow of Time

**DOI:** 10.3390/e24060829

**Published:** 2022-06-14

**Authors:** Snezhana I. Abarzhi, Desmon L. Hill, Annie Naveh, Kurt C. Williams, Cameron E. Wright

**Affiliations:** Faculty of Engineering and Mathematical Sciences, Mathematics and Statistics, The University of Western Australia AUS, Perth 6907, Australia; des.hill@uwa.edu.au (D.L.H.); annie.naveh@gmail.com (A.N.); williamskurt1997@gmail.com (K.C.W.); cwright@aapt.net.au (C.E.W.)

**Keywords:** fluid instabilities, interfacial mixing, blast waves, nuclear synthesis, supernovae, arrow of time

## Abstract

Supernovae are explosions of stars and are a central problem in astrophysics. Rayleigh–Taylor (RT) and Richtmyer–Meshkov (RM) instabilities develop during the star’s explosion and lead to intense interfacial RT/RM mixing of the star materials. We handle the mathematical challenges of the RT/RM problem based on the group theory approach. We directly link the conservation laws governing RT/RM dynamics to the symmetry-based momentum model, derive the model parameters, and find the analytical solutions and characteristics of RT/RM dynamics with variable accelerations in the linear, nonlinear and mixing regimes. The theory outcomes explain the astrophysical observations and yield the design of laboratory experiments. They suggest that supernova evolution is a non-equilibrium process directed by the arrow of time.

## 1. Introduction

Rayleigh–Taylor (RT) and Richtmyer–Meshkov (RM) instabilities and RT/RM interfacial mixing are fluid dynamic phenomena that govern stellar evolution processes and couple astrophysical to atomic scales [1,2,3,4,5,6,7,8,9,10,11,12,13,14,15,16]. Examples include the birth of a star within molecular hydrogen clouds, intense material mixing in stellar interiors, and the death of a star in a supernova (SN) [1,2,3,4,5,6,7,8,9,10]. Supernovae are explosions of stars and are also hypothesized as an astrophysical initial value problem because they may contain information on the entire process of nucleosynthesis and stellar evolution [2,3,4,5,6,7,8,9]. The information is believed to be contained in supernova remnants and in observational data on abundances of chemical elements. Particularly, while light mass elements are produced in a star before its explosion, it is thought that this is the RT/RM mixing of the layers of the exploding star that influences the energy transport, enabling the synthesis of heavy and intermediate mass elements [2,3,4,5,6,7,8,9].

Figure 1 provides a detailed look at the Cassiopeia A (Cas A) supernova remnants produced by the explosion of a massive star [8]. The remnants are seen as broken shells of filamentary structures. The colors mark regions with various chemical compositions. Blue regions contain mostly hydrogen and nitrogen, which exist in the star before the explosion. Bright green regions are rich in oxygen, which is produced primarily during the star’s explosion. Red and purple regions consist mostly of sulfur formed by the oxygen nuclear burning [2,3,4,5,6,7,8,9]. The filamentary structures are generated by the passage of a shock wave from the supernova blast [1,2,5,6]. They are non-uniform, have less mass and move faster than at a speed produced by the star’s free explosion [8]. These filamentary structures are interpreted as due to RT/RM instabilities and RT/RM interfacial mixing developing at the supernova’s blast [1,2,5,6,7].

Stellar environments are associated with large length scales, large velocity magnitudes and small effects of dissipation [1,2,5,6,7]. This suggests high values of Reynolds numbers and may lead to the development of turbulence [1,17]. Canonical turbulence is an essentially isotropic, homogeneous and stochastic process that is free from any memory of deterministic conditions [18,19]. A question thus arises: Do we have, from hydrodynamic perspectives, a direction of time and a principal opportunity to work backward from the observational data in supernova remnants toward the supernova blast and to gain insights beyond the traditional stellar evolution theory [1]?

The asymmetry of time is traditionally associated with the increasing trend for entropy and is explained by the asymmetry of temporal boundary conditions imposed on the Universe at the time of the Big Bang [20]. Active discussions still pertained to whether the direct influence of these initial conditions suffices to explain the broad range of observations with time-directional phenomena at the quantum, classical and relativistic scales. For details see [20,21,22,23,24,25]. We appreciate the importance of the fundamental studies of the arrow of time [20,21,22,23,24,25]. We also acknowledge that stellar evolution and supernovae blasts are multi-scale and multi-physics phenomena with complex interplays of processes and scales, including electro-dynamic, relativistic, quantum, thermodynamic and mechanical aspects [2].

In this work, we consider the fluid dynamics aspects of the titanic problem of supernovae blasts [1]. We focus on the fundamental properties of RT/RM instabilities and RT/RM interfacial mixing with variable accelerations common in supernovae and in high energy density environments [1,12,13,14,15,17,26,27,28,29,30,31,32,33,34,35]. We analyze whether the scale-dependent and scale-invariant RT/RM dynamics depend on the deterministic (the initial and the flow) conditions [1,36]. We intend to gain an understanding from the fluid dynamics perspectives [1]: Can we consider a supernova evolution as a non-equilibrium process directed by the arrow of time?

RT/RM instabilities develop when fluids of different densities are accelerated against their density gradients and exhibit similar features of evolution in vastly different physical regimes [12,13,14,15,37]. In RT/RM dynamics, the growth of small initial perturbations is followed by the interface transforming to the composition of a large-scale coherent structure with the length-scale set by the initial conditions and small-scale shear-driven irregular vortical structures, and then by a transition from the scale-dependent to scale-invariant dynamics with a self-similarly growing length scale in the acceleration direction [37,38]. Accelerations with power-law time dependence characterize supernova environments and yield special scaling properties; they can also be applied for data adjustment in practice [1,31,32,33,34].

While RT/RM instabilities and RT/RM interfacial mixing are challenging to study, significant success was achieved in the understanding of RT/RM dynamics in theory, experiments and simulations [10,11,12,13,14,15,16,17,26,27,28,29,30,35,36,37,38,39,40,41,42,43,44,45,46,47,48,49,50,51,52,53,54,55,56,57]. Particularly, the group theory approach found the order in self-similar RT mixing and explained the high Reynolds number experiments [1,26,27,29,38,40,41,42]. There is a need for a lucid and rigorous theory to treat the problem in its complexity, rationalize the design of experiments, and yield results applicable in a broad astrophysical context, including filaments’ dynamics and energy transport in supernovae [1,17,26,27,28,29,30,37,38,39,40,41,42,43,44].

Here, we directly link the conservation laws governing RT/RM dynamics to the symmetry-based momentum model by applying the group theory approach. We precisely derive the model parameters—buoyancy and drag parameters—in the linear and nonlinear regimes and reveal the mechanisms of transition to self-similar mixing [1,26,27,37,45,51,52,53,54,55,56,57]. We find that while for RT/RM mixing with variable accelerations, the self-similar dynamics can vary from super-ballistics to sub-diffusion, the RT/RM mixing retains memory of deterministic conditions for any acceleration [19]. The sensitivity of the scale-dependent and self-similar RT/RM dynamics to the deterministic and initial conditions suggests that supernova evolution is deterministic in nature and is a non-equilibrium process that follows the arrow of time. We briefly compare our results with astrophysical observations and with laboratory experiments, achieving good agreement. We discuss the experimental design required for accurate implementation, diagnostics and quantification of RT/RM dynamics in supernovae relevant conditions [1,17,26,27,28,29,30,38,39,40,41,42,43,44].

## 2. Fluid Instabilities and Interfacial Mixing

Equations governing RT/RM dynamics involve the equations in the bulk, the boundary value problem and the initial value problem [1,18,37,45,50,51,52,53,54]. In this section, for the reader’s convenience and for the systematicity and rigor, we formulate the governing equations and outline the methods of their solution.

RT/RM dynamics of ideal fluids is governed in the bulk of each fluid by the conservation of mass, momentum and energy
(1)$$\frac{\partial \rho }{\partial t}+\frac{\partial \rho v_{i}}{\partial x_{i}}=0,\;\frac{\partial \rho v_{i}}{\partial t}+\frac{\partial \rho v_{i}v_{j}}{\partial x_{j}}+\frac{\partial P}{\partial x_{i}}=0,\;\frac{\partial E}{\partial t}+\frac{\partial \left(E+P\right)v_{i}}{\partial x_{i}}=0$$
with spatial coordinates $\left(x_{1},x_{2},x_{3}\right)=\left(x,y,z\right)$, time t, and the fields of density, velocity, pressure and energy density $\left(\rho ,v,P,E\right)$, with $E=\rho \left(e+v^{2}/2\right)$, internal energy e and specific enthalpy W=e+P/ρ [1,37]. The closure equation of state relates the pressure and the internal energy; for ideal fluids P=s ρ e with some constant s. These equations are augmented with the boundary conditions at the interface and the boundaries of the outside domain
(2)$$\begin{array}{r} \left[\rho \left(\frac{1}{\left|\nabla \theta \right|}\frac{\partial \theta }{\partial t}+v\cdot n\right)\right]=0,\;\left[v\cdot n\right]=0,\;\left[v\cdot \tau \right]=arbitrary,\;\left[P\right]=0,\;\left[W\right]=arbitrary,\\ {\left.v\right|}_{z\to +\infty }=0,\;{\left.v\right|}_{z\to -\infty }=0 \end{array}$$
where $\left[\ldots \right]$ denotes the quantity jump across the interface, $n\;\left(\tau \right)$ is the normal (tangential) unit vector of the interface defined as $n=\nabla \theta /\left|\nabla \theta \right|,\;\left(n\cdot \tau \right)=0$, with the function $\theta \left(x,y,z,t\right)=0$ at the interface and with $\theta >0\;\left(<0\right)$ in the bulk of the heavy (light) fluid marked with sub-script $h\left(l\right)$. The initial conditions are the initial perturbations of the interface and the flow fields. They define the flow symmetry and the characteristic scales. We consider the dynamics of a spatially extended flow periodic in the $\left(x,y\right)$ plane, as set by the initial conditions. RTI/RMI is driven by acceleration g directed from the heavy to the light fluid along the z axis, $g=\left(0,0,-g\right)$, $g=\left|g\right|$ [1,12,13,14,15,26]. The acceleration modifies the pressure P→P−ρgz, and is a power-law function of time, $g=Gt^{a}$, $t>t_{0}>0$, with exponent $a\in \left(-\infty ,+\infty \right)$ and strength G>0. The Atwood number $A=\left(\rho_{h}-\rho_{l}\right)/\left(\rho_{h}+\rho_{l}\right)$ parameterizes the fluids’ density ratio [1,37,52]. For non-ideal fluids, the governing equations are further modified; in particular, the presence of the kinematic viscosity ν augments the momentum equation with the term $\left(-\rho \nu \partial^{2}v_{i}/\partial x_{j}^{2}\right)$ [18].

Under conditions of high energy density relevant to supernovae, RT/RM dynamics are usually driven by strong variable shocks [1,17,26,27,28,29]. The post-shock dynamics is a superposition of two motions—the background motion of the fluid bulk and the growth of the interface perturbations. In the background motion, both fluids (in bulk between the reflected and transmitted shock) and their interface move as a whole unit in the direction of the transmitted shock. This motion occurs even for an ideally planar interface; it is supersonic and even hypersonic for strong shocks. The RT/RM growth of the interface perturbations is due to shock-induced acceleration. It develops only for a perturbed interface or flow field. The rate of this growth is sub-sonic and the associated RT/RM motion is nearly incompressible [1,26,29,30,49]. Besides, RT/RM dynamics is essentially interfacial—with intense fluid motion near the interface, with effectively no fluid motion away from the interface, and with shear-driven vortical structures at the interface [1,12,13,14,15,26,37,38,40,45,49,50,51,52,53,54]. 

The RT/RM unstable interface is composed of a large-scale coherent structure and a small-scale—usually irregular—structures [1,37,38]. For the large-scale structure of bubbles and spikes, where the bubble (spike) is the portion of the light (heavy) fluid moving up (down) into the heavy (light) fluid, the velocity field is potential in the bulk [37,45,52]. Small-scale vortical structures appear at the interface due to shear and Kelvin–Helmholtz instability caused by disjointed tangential velocity components and enthalpy at the interface [17,18,26,27,28,29,30,37,38,39,40,45]. The RT/RM coherent structure can be viewed as a standing wave with a growing amplitude [1,37,52]. As time evolves, the interaction of scales enhances, and the dynamics transit from the scale-dependent to the scale-invariant regime [1,37].

The mathematical problem of RTI/RMI requires one to solve the system of nonlinear partial differential equations in four-dimensional space-time, solve the boundary value problem at the unstable nonlinear interface and the outside boundaries, and also solve the ill-posed initial value problem with an account for non-locality and singularities Equation (2) [1,37,52]. Despite this extreme complexity, RT/RM dynamics are observed to have certain features of universality and order, and can be treated from the first principles by applying group theory [17,26,27,28,29,30,37,38,39,40,41,42,43,44,45]. For scale-dependent dynamics, group theory can employ space groups to derive the dynamical system from the governing equations and find its asymptotic solutions. For scale-dependent and self-similar dynamics, group theory can be realized in the momentum model having the same symmetries and scaling transformation as the governing equations [1,26,37,38,40,41,42,43,44,45,51,52,53,54,55].

Can we directly link the two group theory realizations in a lucid yet rigorous theory to efficiently handle the mathematical challenges of the RT/RM problem? What can the unified framework tell us about the scale-dependent and self-similar RT/RM dynamics? What are the theory outcomes for supernova evolution and the arrow of time, for the characteristics of filamentary structures in supernova remnants, and for the design of laboratory experiments? These questions motivate and frame our study.

## 3. Results

### 3.1. Group Theory

For spatially periodic flows, the scale-dependent RT/RM dynamics is invariant with respect to a space group whose generators are translations in the plane, rotations and reflections, and which also has anisotropy in the acceleration direction and inversion in the normal plane, such as the hexagonal or square groups in a three-dimensional (3D) flow [1,37,45]. By considering incompressible large-scale coherent dynamics with potential velocity field(s) $v_{h\left(l\right)}=\nabla \Phi_{h\left(l\right)}$, and by applying irreducible representations of the relevant group, we expand the flow fields as Fourier series and make the spatial expansion in the vicinity of a regular point of the interface—the tip of the bubble or spike with coordinate $\left(0,0,Z\left(t\right)\right)$ and velocity $\left(0,0,v\left(t\right)\right)$ with $v\left(t\right)=\dot{Z}$ (dot marks partial time-derivative). This reduces the governing equations in Equations (1) and (2) to a dynamic system for moments (which are infinite sums of Fourier amplitudes) and surface variables [1,37,42,45,51,52]. For the hexagonal group, to the first order, N=1, the interface is $z^{*}-Z\left(t\right)=\zeta \left(x^{2}+y^{2}\right)$, and the dynamical system is [1,37,51,52]:(3)$$\begin{array}{c} \rho_{h}\;\left(\dot{\zeta }-2\zeta M_{1}-\frac{M_{2}}{4}\right)=0,\;\rho_{l}\left(\dot{\zeta }-2\zeta {\tilde{M}}_{1}+\frac{{\tilde{M}}_{2}}{4}\right)=0,\;M_{1}-{\tilde{M}}_{1}=arbitrary,\\ \rho_{h}\;\left(\frac{{\dot{M}}_{1}}{4}+\zeta {\dot{M}}_{0}-\frac{M_{1}^{2}}{8}+\zeta g\right)=\rho_{l}\left(\frac{{\dot{\tilde{M}}}_{1}}{4}-\zeta {\dot{\tilde{M}}}_{0}-\frac{{\tilde{M}}_{1}^{2}}{8}+\zeta g\right),\;M_{0}=-{\tilde{M}}_{0}=-v \end{array}$$
Here $M\;\left(\tilde{M}\right)$ are the moments of the heavy (light) fluid, and $v\geq 0\;\left(\leq 0\right),\;\zeta \leq 0\;\left(\geq 0\right)$ are the velocity and the curvature of the interface at the tip of the bubble (spike). The system length-scale is $k^{-1}$, where the wavevector is $k\sim \lambda^{-1}$ and the wavelength is λ. The time-scale is $\tau =\tau_{G}={\left(kG\right)}^{-1/\left(a+2\right)}$ for acceleration driven RT type dynamics with a>−2; it is $\tau =\tau_{0}={\left(kv_{0}\right)}^{-1}$ for initial growth-rate driven RM type dynamics with a<−2, where $v_{0}=\left|v\left(t_{0}\right)\right|$ is the initial growth rate of the interface initial perturbation with $\left|\zeta_{0}k\right|<<1,\;\zeta_{0}=\zeta \left(t_{0}\right)$. Group theory is further applied to solve the closure problem and to find linear and nonlinear solutions for scale-dependent RT/RM dynamics [1,37,42,45,51,52].

Alternatively, by analyzing symmetries and scaling transformations of the governing equations, we can theorize that in RT/RM flow the dynamics of a parcel of fluid (in the bubble or spike region) is governed by a balance per unit mass of the rates of momentum gain, $\tilde{\mu }$, and momentum loss, μ, as
(4)$$\dot{h}=v,\;\dot{v}=\tilde{\mu }-\mu$$
where h is the length scale along the acceleration g, v is the corresponding velocity, $\tilde{\mu }\left(\mu \right)$ is the magnitude of the rate of gain (loss) of specific momentum in the acceleration direction. The rate of gain (loss) of a specific momentum is $\tilde{\mu }=\tilde{\varepsilon }/v$ (μ=ε/v), with $\tilde{\varepsilon }\left(\varepsilon \right)$ being the rate of gain (dissipation) of specific energy. The rates of energy gain and dissipation are $\tilde{\varepsilon }=Bgv$ and $\varepsilon =Cv^{3}/L$, with B and C being the buoyancy and drag parameters, respectively. The length scale L for energy dissipation can be the horizontal scale (wavelength) $L\sim \lambda \sim k^{-1}$, or the vertical scale (amplitude) $L\sim \left|h\right|$. The case L~λ corresponds to scale-dependent linear and nonlinear dynamics, whereas the case L~h correspond to scale-invariant mixing dynamics [1,26,37,53,54]. This summarizes to:(5)$$\tilde{\mu }=\frac{\tilde{\varepsilon }}{v},\;\mu =\frac{\varepsilon }{v},\;\tilde{\varepsilon }=Bgv,\;\varepsilon =C\frac{v^{3}}{L},\;L\sim \left\{\;\lambda \;,\;\lambda \;,\;\left|h\right|\;\right\}$$

One can find solutions for the momentum model in the linear, nonlinear and mixing regimes. The asymptotic solutions reveal that the dynamics is RT (RM) type for $a>-2\;\left(<-2\right)$ in the scale-dependent linear and nonlinear regimes and for $a>a_{cr}\;\left(<a_{cr}\right)$ in the scale-invariant regime, $a_{cr}=-2+{\left(1+C\right)}^{-1}$ [1,19].

An idea to describe RT/RM dynamics through buoyancy and drag balance has been discussed in the RT/RM research field for a long time [48,56,57,58,59]. The group theory-based momentum model reconciles with these studies and provides a number of important advantages [1,36,54,58]. (1) The momentum model represents RT/RM dynamics by considering balances per unit mass (rather than per unit volume). This specific, per unit mass, dynamics of RT/RM mixing is due to the independence of the boundary condition $\left[v\cdot n\right]=0$ on the fluid density $\rho_{h\left(l\right)}$. (2) The momentum model has the same symmetries and scaling transformations as the governing equations. (3) The momentum model captures the physics of RT/RM dynamics by linking the specific rates of change of momentum and the specific rates of change of energy as $\tilde{\varepsilon }=\tilde{\mu }\;v$ and ε=μ v. See Equations (1)–(4) [1,36,54,58]. A comparative study of the momentum model, the interpolation models, turbulence models and other approaches in RT/RM mixing is given in [58].

In the present work, we directly link the dynamical system, the momentum model and the governing equations in Equations (1)–(5), derive the model parameters and solve the model equations in RT/RM linear, nonlinear and mixing regimes. This elaborates the rigorous theoretical framework for studying RT/RM dynamics in a broad range of conditions, identifies the sensitivity of non-equilibrium RT/RM dynamics to deterministic conditions, and illustrates that supernovae are indeed an astrophysical initial value problem with the directed arrow of time.

### 3.2. Scale-Dependent Linear Dynamics

In the dynamical system, we associate the curvature ζ with the amplitude Z, kZ=−4ζ/k, and retain only first-order harmonics in the expressions for the moments. In the momentum model, we relate the vertical scale with the amplitude, h=Z, and the length scale with the wavevector, $L=k^{-1}$ [1,37,52,54]. This transforms the governing equations in Equations (3)–(5) to
(6)$$\dot{Z}=v,\;\dot{v}=B_{l}g-C_{l}kv^{2},\;\zeta =-\frac{k^{2}Z}{4},\;\zeta =\xi k,\;v=-M_{0}={\tilde{M}}_{0}$$
where the buoyancy and the drag parameters are:(7)$$B_{l}=AkZ=-4A\frac{\zeta }{k},\;C_{l}=\frac{A}{2}$$
The sub-script l stands for linear. In the linear regime, the buoyancy parameter is time-dependent, with $B_{l}>0\;\left(<0\right)$ for the bubble (spike), the drag parameter is constant $C_{l}>0$, and they both depend on A. Table 1 summarizes these results.

By integrating these equations in Equation (6) for $g=Gt^{a}$, we obtain solutions for the early-time RT dynamics with a>−2 and for the early-time RM dynamics with a<−2 for bubbles (spikes), which move up (down) and are concave down (up) with Z>0, v>0, ζ<0 $\left(<0,\;<0,\;>0\right)$:(8)$$\begin{array}{l} a>-2:\;Z=c_{\pm }\;\;\frac{1}{k}{\left(\frac{t}{\tau_{G}}\right)}^{1/2}\;I_{\pm 1/2s}\left(\frac{\sqrt{A}}{s}{\left(\frac{t}{\tau_{G}}\right)}^{a}\right),\;v=\dot{Z},\;\zeta =-\frac{Zk^{2}}{4}\;;\\ a<-2:\;Z=\;\frac{1}{2A\;k}ln\left(c_{+}\left(\frac{t}{\tau_{0}}\right)+c_{-}\right),\;v=\dot{Z},\;\zeta =-\frac{Zk^{2}}{4} \end{array}$$
Here $s=\left(a+2\right)/2$, $I_{p}$ is the modified Bessel function of the p th order and $c_{\pm }$ are the integration constants. The formation of bubbles and spikes is defined by the initial conditions, with an RT bubble (spike) developing for $\zeta \left(t_{0}\right)<0\;\left(>0\right)$ and with RM bubble (spike) developing for $v\left(t_{0}\right)>0\;\left(<0\right)$. For RT/RM bubble/spike the curvature magnitude increases with time, whereas the velocity magnitude increases (decreases) in the RT (RM) case. In the linear regime, the RT-RM transition occurs at a=−2 upon the by varying the acceleration strength Gk [1,52]. In the early-time linear regime, the effect of deterministic conditions is revealed in the dependence on the initial conditions of the RT/RM growth and growth rates.

As time progresses, the bubble/spike amplitude increases, its curvature approaches a constant value, and RT/RM dynamics become nonlinear [1].

### 3.3. Scale-Dependent Nonlinear Dynamics

In the dynamical system, we impose the constant curvature condition and retain higher-order harmonics in the moments. In the momentum model, we relate the vertical scale with the amplitude, $h=z_{0}$, and the length scale with the wavevector, $L=k^{-1}$ [1,37,52,54]. This transforms the governing equations in Equations (3)–(5) to
(9)$${\dot{z}}_{0}=v,\;\dot{v}=B_{n}g-C_{n}kv^{2},\;\zeta =\xi k,\;v=-M_{0}={\tilde{M}}_{0}$$

In the nonlinear regime, the buoyancy and the drag parameters are:(10)$$B_{n}\left(A,\xi \right)=-2A\xi \frac{\left(9-64\xi^{2}\right)}{\left(3+10A\xi -128A\xi^{3}\right)},\;C_{n}\left(A,\xi \right)=\frac{\left(9A-48\xi +64A\xi^{2}\right)}{\left(9-64\xi^{2}\right)\left(3+10A\xi -128A\xi^{3}\right)}$$
in the domain $\xi \in \left[{\overset{⌢}{\xi }}_{cr},\;0\right]\cup \;\left[\xi_{A},\;{\overset{⌣}{\xi }}_{cr}\right]$ with ${\overset{⌣}{\xi }}_{cr}=-{\overset{⌢}{\xi }}_{cr}=3/8$. The sub-script n stands for nonlinear, and $\overset{⌢}{\ldots }\left(\overset{⌣}{\ldots }\right)$ are used to mark bubbles (spikes). For nonlinear RT/RM bubbles, the values of the buoyancy and drag parameters are $B_{n}\geq 0,\;C_{n}\geq 0$ for $\xi \in \left(-\xi_{cr},0\right)$ with $B_{n}\in \left[0,{\overset{⌢}{B}}^{*}\right],\;C_{n}\in \left[{\overset{⌢}{C}}^{*},\infty \right)$. For nonlinear RT/RM spikes, the values are $B_{n}\leq 0$ with range $\left[{\overset{⌣}{B}}^{*},0\right]$ for $\xi \in \left(0,\xi_{cr}\right)$, and $C_{n}\leq 0$ with range $\left(-\infty ,0\right]$ for $\xi \in \left({\overset{⌣}{\xi }}_{A},{\overset{⌣}{\xi }}_{cr}\right)$. Here ${\overset{⌣}{\xi }}_{A}={\overset{⌣}{\xi }}_{A}\left(A\right)$ is the curvature of the fastest spike with ${\overset{⌣}{\xi }}_{A}=3\left(1-\sqrt{1-A^{2}}\right)/8A$ and with ${\overset{⌣}{\xi }}_{A}\in \left(0,{\overset{⌣}{\xi }}_{cr}\right)$. We call it the Atwood spike.

Table 2 summarizes the results. Figure 2a,b present the magnitudes of the buoyancy and drag parameters for bubbles and spikes as functions on the interface morphology—the dimensionless curvature ξ=ζ/k for some Atwood numbers in the nonlinear regime. Overall, the buoyancy and drag magnitudes are greater for bubbles than for spikes and are also greater for greater A.

Hence, we find that for given A in nonlinear RT/RM dynamics there is a family of buoyancy and drag values parameterized by the interface morphology, i.e., the curvature of the bubble/spike ζ. In the family, for given $\left(A,\zeta \right)$ the values $\left(B_{n},C_{n}\right)$ are constant. Non-uniqueness is due to the presence of shear at the interface [1].

We integrate the system of equations in Equations (9) and (10) for $g=Gt^{a}$ to obtain solutions describing the RT/RM dynamics of nonlinear bubbles/spikes. The second equation is a Riccati nonlinear differential equation. For $g=Gt^{a}$ it is transformed to a linear equation $U_{TT}-Q^{2}T^{a}U=0$. For RT dynamics with a>−2 these quantities are $Q^{2}=Q_{RT}^{2}=B_{n}C_{n}$ with $V=U_{T}/UC_{n}\left(\tau_{G}/\tau_{0}\right)$, $V=v/v_{0}$ and $T=t/\tau_{G}$. For RM dynamics with a<−2 these quantities are $Q^{2}=Q_{RM}^{2}={\left(\tau_{0}/\tau_{G}\right)}^{2s}B_{n}C_{n}$ with $V=U_{T}/UC_{n}$, $V=v/v_{0}$, $T=t/\tau_{0}$ and $s=\left(a+2\right)/2$. The solution is presented in terms of modified Bessel functions as
(11)$$U=c_{\pm }\;T^{1/2}\;I_{\pm 1/2s}\left(\frac{QT^{s}}{s}\right)$$
By further applying to this explicit solution in Equation (11) the Taylor series expansions, we find nonlinear asymptotic solutions for a>−2 with $T^{s}\to \infty$ for T→∞, and for a<−2 with $T^{s}\to 0$ for T→∞ as:(12)$$\begin{array}{l} a>-2:\;\frac{t}{\tau_{G}}\to \infty ,\;{\left(\frac{t}{\tau_{G}}\right)}^{\left(a+2\right)/2}\to \infty ,\;v\to v_{RT},\;v_{RT}=\pm \sqrt{\frac{Gt^{a}}{k}}\sqrt{\frac{B_{n}}{C_{n}}}\;;\\ a<-2:\;\frac{t}{\tau_{0}}\to \infty ,\;{\left(\frac{t}{\tau_{0}}\right)}^{\left(a+2\right)/2}\to 0,\;v\to v_{RM},\;v_{RM}=\frac{1}{kC_{n}t}. \end{array}$$
We emphasize, see Equations (11) and (12) that nonlinear RT dynamics are achieved for $\left(t/\tau_{G}\right)\to \infty$ and ${\left(t/\tau_{G}\right)}^{\left(a+2\right)/2}\to \infty$, whereas nonlinear RM dynamics are achieved for $\left(t/\tau_{0}\right)\to \infty$ and ${\left(t/\tau_{0}\right)}^{\left(a+2\right)/2}\to 0$ akin to climbing up descending stairs. In the nonlinear linear regime, the RT-RM transition occurs at a=−2 upon varying of the acceleration strength Gk [1,52]. In the nonlinear regime, the effect of deterministic conditions is revealed in the dependence of RT/RM bubble/spike dynamics on the initial conditions, including symmetry and length scale. Figure 3a,b presents solutions for RT/RM bubbles and spikes for some Atwood numbers in the nonlinear regime.

Note the presence of special solutions in RT/RM families (to be discussed in detail in the future). These include the flat bubble/spike with zero curvature ${\overset{⌣}{\xi }}_{f}={\overset{⌢}{\xi }}_{f}=0$; the critical bubble/spike having the largest curvature magnitude ${\overset{⌣}{\xi }}_{cr}=-{\overset{⌢}{\xi }}_{cr}=3/8$; the Taylor bubble/spike having the curvature magnitudes ${\overset{⌣}{\xi }}_{T}=-{\overset{⌢}{\xi }}_{T}=3/8$ similarly to the bubble observed by Davies & Taylor [13]; the Layzer-drag bubble and spike with the velocity dependent on the Atwood number A as ${\overset{⌢}{v}}_{LD}/\sqrt{g/k}=\sqrt{2A/\left(1+A\right)}$ and ${\overset{⌣}{v}}_{LD}/\sqrt{g/k}=\sqrt{2A/\left(1-A\right)}$ [52,56]; the fastest Atwood bubble with ${\left({\overset{⌢}{\zeta }}_{A},{\overset{⌢}{v}}_{A}\right)}_{\mathrm{RT}/\mathrm{RM}}$ and the fastest Atwood spike with ${\left({\overset{⌣}{\zeta }}_{A},{\overset{⌣}{v}}_{A}\right)}_{\mathrm{RT}/\mathrm{RM}}$ [1,52].

RT dynamics are set by the interplay of buoyancy and drag depending on the interface morphology and shear. The dynamics of RT bubbles is regular and is influenced by the following competing factors: more curved bubbles have larger buoyancy and move faster than flattened bubbles; yet, bubbles with larger curvature have larger shear and larger drag reducing their velocities; for curved bubbles, the shear alone can maintain the pressure at the interface leading to zero buoyancy and infinite drag. The fastest Atwood bubble has the invariance ${\left[{\overset{⌢}{v}}_{A}\tau k\;{\left(t/\tau_{G}\right)}^{-a/2}{\left(8\left|{\overset{⌢}{\zeta }}_{A}\right|/k\right)}^{-3/2}\right]}_{\mathrm{RT}}=1$. The dynamics of RT spikes are singular. While the magnitude of the spike’s buoyancy is qualitatively similar to that of the bubble, the spike’s drag vanishes $C_{n}\to 0$ for $\xi \to {\overset{⌣}{\xi }}_{A}$. This leads to a singularity, indicating that the RT Atwood spike has velocity growing quickly with time, ${\left[\left|{\overset{⌣}{v}}_{A}\right|/\sqrt{Gt^{a}/k}\right]}_{\mathrm{RT}}\to \infty$. See Figure 3a.

RM dynamics are set only by the drag depending on the interface morphology and shear. For regular RM bubbles, the drag is minimal for ξ→0 and approaches infinity for $\xi \to \xi_{cr}$, indicating that in nonlinear RMI, the fastest Atwood bubble is the flat bubble ${\overset{⌢}{\xi }}_{A}={\overset{⌢}{\xi }}_{f}=0$; it has the quasi-invariance ${\left[\left(4/3\right)tv_{A}^{2}/{\left.|dv/d\zeta \right|}_{\zeta =\zeta_{A}}|\right]}_{\mathrm{RM}}\approx 1$. The dynamics of RM spikes is singular, because the spike’s drag vanishes $C_{n}\to 0$ for $\xi \to {\overset{⌣}{\xi }}_{A}$, which is the same as in the RT case. This leads to a singularity, suggesting that the RM Atwood spike has the velocity and shear growing quickly with time, ${\left[|{\overset{⌣}{v}}_{A}|kt\right]}_{\mathrm{RM}}\to \infty$. See Figure 3b.

### 3.4. Transition to Self-Similar Mixing

Linear RT/RM dynamics depend on the horizontal length scale—the wavelength λ. Linear RT/RM dynamics develop faster for smaller wavelengths $\lambda \sim k^{-1}$ as is set by times-scales, $\tau_{G}\sim {\left(\lambda /G\right)}^{\left(a+2\right)}$ and $\tau_{0}\sim \lambda /v_{0}$. Nonlinear RT/RM dynamics depend on the vertical and horizontal length scales—the wavelength λ and the amplitude h. Nonlinear RT/RM dynamics are faster for larger wavelengths, as dictated by the bubble/spike velocities, $\left|v_{RT}\right|\sim \sqrt{\lambda Gt^{a}}$ and $\left|v_{RM}\right|\sim \lambda /t$. With time, RT/RM dynamics transit from the scale-dependent nonlinear regime to the self-similar mixing. The transition can occur due to the bubbles merging and the spikes merging and the multi-pole interactions caused by the growth of the horizontal scale [55,56,57]. Discussions of the importance of the merger mechanism and the classification of the ‘merger’ transitions in RT/RM structures that excellently agree with experiments are given in [35,46,47,48,55,56,57]. Besides, RT/RM dynamics can become self-similar when the amplitude is the dominant scale for energy dissipation [53,54]. The first result on the amplitude dominance mechanism is reported in [54].

Indeed, near the tip of the bubble/spike, in the laboratory frame of reference, the interface is described as $z^{*}=Z+\zeta \left(x^{2}+y^{2}\right)$, with the velocity $v=\dot{Z}$ and the amplitude Z, with $\left|\zeta \right|\sim \lambda^{-1}$ and $\left|x\right|,\left|y\right|\sim \lambda$. In nonlinear RT dynamics their values are $\left|v\right|\sim \sqrt{\lambda Gt^{a}}$ and $\left|z_{0}\right|\sim t\sqrt{\lambda Gt^{a}}$. In nonlinear RM dynamics their values are $\left|v\right|\sim \lambda /t$ and $\left|z_{0}\right|\sim \lambda ln\left(v_{0}t/\lambda \right)$. For the fastest Atwood bubble, the dominance of the amplitude, with $\left|z^{*}\right|\sim \left|Z\right|$ for $\left|z_{0}\right|>\lambda$, may reduce the drag force from $\sim \left|v^{2}/\lambda \right|$ to $\sim \left|v^{2}/Z\right|$ and accelerate RT/RM bubbles. For the fastest Atwood spike, the drag is zero $C_{n}\to 0$, thus leading to the acceleration of RT/RM spikes.

The traditional merge mechanism and the amplitude dominance mechanism both can lead to the acceleration of RT/RM bubbles/spikes to due to the drag reduction, and can transfer the scale-dependent dynamics with L~λ to the scale-invariant mixing with $L\sim \left|Z\right|$, in agreement with high Reynolds number experiments [28,29,38]. 

### 3.5. Self-Similar Mixing

In the dynamical system, we generalize the equations in Equations (1)–(5) as
(13)$$\dot{Z}=v,\;\dot{v}=B_{m}g\;-\;C_{m}\frac{v^{2}}{\left|Z\right|}$$
The sub-script m stands for mixing. Here, the buoyancy and the drag parameters are $B_{m},C_{m}>0\;\left(<0\right)$ for RT/RM bubble (spike) with $Z,v>0\;\left(<0\right)$. The buoyancy and drag are now free parameters due to the many scales contributing; they can also be stochastic processes due to the randomness of mixing [1]. Table 3 summarizes the results. In the momentum model, we relate the length-scale for energy dissipation with the amplitude $L=\left|h\right|=\left|Z\right|$; consider for definiteness $B_{m},C_{m}>0$; and we re-scale the values as $B_{m}Gt^{a}\to Gt^{a},\;C_{m}\to C$ with $C\in \left(0,\infty \right)$. We find solutions in the domain $h,v,t>{\tilde{t}}_{0}$ with some initial instance of time ${\tilde{t}}_{0}>>t_{0}>0$ for the equation in Equation (13) representing the dynamics of a fluid parcel undergoing RT/RM mixing [1]:(14)$$\overset{‥}{h}+C\frac{{\dot{h}}^{2}}{h}-Gt^{a}=0$$

The non-homogeneous equation in Equation (14) has particular solution $h=h_{RT}$, which we find by applying the Lie groups to be $h_{RT}=B_{RT}t^{b_{RT}}$. It describes self-similar RT type mixing. The associated homogeneous equation has general solution $h^{C+1}=H_{0}^{C+1}+H_{0}^{C}\left(1+C\right)V_{0}\left(t-{\tilde{t}}_{0}\right)$ with integration constants $V_{0},H_{0}$, and with $h\to h_{RM}$ for $t/{\tilde{t}}_{0}\to \infty$. This asymptotic solution describes self-similar RM type mixing. The momentum model couples general and particular solutions. In asymptotic limits, the solutions $h_{RT/RM}$ are effectively decoupled due to their distinct symmetries, which are the scaling symmetry and the point group, respectively. This leads to:(15)$$\begin{array}{l} a>a_{cr}:\;h\to h_{RT},\;h_{RT}=B_{RT}t^{b_{RT}},\;b_{RT}=a+2,\;B_{RT}=\frac{G}{\left(a+2\right)\left(\left(a+1\right)+C\left(a+2\right)\right)};\\ a<a_{cr}:\;h\to h_{RM},\;h_{RM}=B_{RM}t^{b_{RM}},\;b_{RM}=a_{cr}+2,\;B_{RM}=H_{0}{\left(\left(1+C\right)\left(\frac{V_{0}}{H_{0}}\right)\right)}^{1/\left(1+C\right)}. \end{array}$$
In the self-similar mixing regime, the RT-RM transition occurs for $a\sim a_{cr}$, where the critical exponent is $a_{cr}=-2+{\left(1+C\right)}^{-1}$ with $a_{cr}\in \left(-2,-1\right)$ for $C\in \left(0,\;+\infty \right)$ [1,29,36].

For the solution $h_{RT}$ describing self-similar RT mixing, the power-law exponent is set by the acceleration exponent, the pre-factor is set by the acceleration parameters and drag, and the rates of change of momentum relate as $\left|\tilde{\mu }\right|\sim \left|\mu \right|\sim t^{a}$. For the solution $h_{RM}$ describing self-similar RM-type mixing, the power-law exponent is set by the drag, the pre-factor is set by deterministic conditions, and the rates of change of momentum relate as $\left|\mu \right|\sim \left|\dot{v}\right|\sim t^{a_{cr}}$. RM mixing is faster than the acceleration prescribes.

### 3.6. Properties of Self-Similar Mixing

In RT mixing with $a>a_{cr}$, the length and velocity scale with time as $L\sim t^{a+2}$ and $v\sim t^{a+1}$ the velocity scales with length as $v\sim L^{\left(a+1\right)/\left(a+2\right)}$. While the length scale L increases with time for with $a>a_{cr}$, the velocity scale increases (decreases) with time for $a>-1\;\left(<-1\right)$ with larger velocities corresponding to larger (smaller) length scales for $a>-1\;\left(<-1\right)$; it is constant at a=−1. In RM mixing with $a<a_{cr}$, the length and velocity scale with time as $L\sim t^{a_{cr}+2}$ and as $v\sim t^{a_{cr}+1}$ and the velocity scales with length as $v\sim L^{\left(a_{cr}+1\right)/\left(a_{cr}+2\right)}$. Since $a_{cr}\in \left(-2,-1\right)$, the length scale L increases with time, the velocity scale decreases with time, and larger velocities correspond to smaller length scales. This yields a special self-similar class for RT/RM mixing [1,18,19,36,37,43].

For RT mixing with $a>a_{cr}$ the dynamics is super-ballistic (i.e., faster than ballistics) for a>0 with $v\sim L^{\left(a+1\right)/\left(a+2\right)}$; it is ballistic at a=0 with $L\sim v^{2}$. The dynamics is quasi-Kolmogorov at a=−1/2 with $L\sim v^{3}$. The dynamics is ‘steady flex’ at a=−1 separating the sub-regimes with larger (smaller) velocities associated with larger (smaller) length scales. The dynamics is super-diffusive (i.e., faster than diffusion) for a>−3/2; quasi-diffusive at a=−3/2 with $L\sim v^{-1}$; and sub-diffusive (i.e., slower than diffusion) for $a\in \left(-3/2,\;a_{cr}\right)$. RM mixing for $a<a_{cr}$ has larger velocities at smaller length scales, $v\sim L^{\left(a_{cr}+1\right)/\left(a_{cr}+2\right)}$; its dynamics are sub-diffusive for C>1 [36].

To evaluate the sensitivity of RT/RM mixing to deterministic conditions, we consider two parcels of fluids involved in the RT (RM) mixing flow with a time-delay $\tilde{\tau }$. The relative velocity of the parcels is $\sim {\tilde{\tau }}^{\left(a+1\right)}\;\left({\tilde{\tau }}^{\left(a_{cr}+1\right)}\right)$, the parcel’s velocity is $\sim t^{\left(a+1\right)}\;\left(t^{\left(a_{cr}+1\right)}\right)$, and their ratio is $\sim {\left(\tilde{\tau }/t\right)}^{\left(a+1\right)}\;\left({\left(\tilde{\tau }/t\right)}^{\left(a_{cr}+1\right)}\right)$. This suggests that the effect of deterministic conditions decreases (increases) with time for $a>-1\;\left(<-1\right)$ in RT mixing $a>a_{cr}$, and increases with time in RM mixing $a<a_{cr}$. Hence, for RT/RM mixing with variable acceleration, $g=Gt^{a}$, the self-similar dynamics can vary with the acceleration exponent a from super-ballistics to sub-diffusion, with deterministic conditions having stronger influence for smaller a.

## 4. Outcome of the Theory

### 4.1. Fundamental Aspects

The direct link between the dynamical system and the momentum model demonstrates that these implementations of group theory are fully consistent with one another and with the governing equations, and that the group theory approach provides a rigorous theoretical framework for studying the RT/RM dynamics in a broad range of conditions.

We reveal the fundamental features of RT/RM dynamics. (i) The parameters of the momentum model are derived and the solutions for RT/RM dynamics are found in the linear, nonlinear and mixing regimes. (ii) In the linear regime, the buoyancy magnitude grows with amplitude and time, and the drag is set by the density ratio. (iii) In the nonlinear regime, the buoyancy and the drag parameters depend on the interface morphology and interfacial shear, as well as the density ratio. (iv) In the mixing regime, the buoyancy and the drag are independent free parameters. They can also be stochastic processes. (v) In any regime, RT/RM dynamics are interfacial, with intense fluid motion near the interface, with no motion away from the interface, and with shear-driven vortical structures at the interface [1].

Our results identify important physics properties of RT/RM dynamics. (i) RT/RM dynamics are specific and are driven by balance per unit mass (rather than per unit volume). The specificity is due to the boundary conditions at the interface. (ii) In each regime, the buoyancy and the drag parameters have the same values in the RT and RM cases. The type of the dynamics—RT or RM—and the RT—RM transition is defined by the acceleration. (iii) The buoyancy and the drag parameters are different for bubbles and spikes. For instance, nonlinear bubbles have overall greater buoyancy and drag magnitudes when compared to nonlinear spikes. (iv) The buoyancy and the drag parameters are distinct in different regimes. (v) Non-equilibrium RT/RM dynamics sense deterministic conditions in any—linear, nonlinear, mixing—regime.

### 4.2. Astrophysical Aspects

In supernova blasts, RT/RM instabilities and RT/RM interfacial mixing are induced by the accelerations $g=Gt^{a}$, and the exponents are $a\in \left(-2,-1\right)$ for the first and second-kind self-similarities [1,2,3,4,5,6,7,8,9,10,18,31,32,33,34]. For these exponent values: The dynamics is RT type—acceleration driven—in the linear regime, with the structure of bubbles and spikes set by the interface morphology. It is also an RT type in the nonlinear regime, with buoyancy and drag set by the interfacial shear. Yet, the dynamics is RM type in the self-similar mixing regime and is faster than the acceleration prescribes. For large drag values found in observations [17,26,27,28,29,30,34,35,38,39,40,46,47,48,49,56,57], the self-similar RM mixing is a slow sub-diffusive process, with strong dependence on deterministic conditions, with smaller velocities at larger length scales, and with non-uniformities and localizations.

Our analysis is consistent with and explains the observations of filamentary structures in supernova remnants. These structures are due to RT/RM instabilities and RT/RM interfacial mixing developing at the supernova blast. They are non-uniform, have less mass and move at a speed higher than the free explosion of a star can produce, in conformity with our results (Figure 1). The sub-diffusive character of self-similar mixing found in our work suggests a special mechanism for energy transport at microscopic scales—through energy trapping and localizations—to enable nucleosynthesis of heavy and intermediate mass elements in supernovae conditions. The sensitivity of RT/RM dynamics to deterministic conditions in scale-dependent and self-similar regimes implies that supernova evolution is a non-equilibrium process directed by the arrow of time. The information on the star’s structure at the time of the explosion is encapsulated in and can be deduced from observational data in supernova remnants. We find that supernova evolution is indeed the astrophysical initial value problem [1].

### 4.3. Scaled Laboratory Experiments

Conditions of high energy density (HED) in supernovae can be realized at high-power laser facilities [17,26,27,28,29,30]. This enables studies of cosmic events and explosion processes in the laboratory and links astrophysics to plasma fusion [17,26,27,28,29,30]. To gain a deeper understanding of supernova remnants and to achieve better control of plasma fusion, RT/RM dynamics can be modeled in scaled laboratory experiments for broad sets of acceleration patterns [1,2,3,4,5,6,7,8,9,17,26,27,28,29,30,38,39,40,41,42,43,44].

Since linear and nonlinear RT/RM dynamics are described by standardized functions, one can link the experimental results obtained for increasing with time accelerations a>0 to supernova relevant cases −2<a<0, and vice versa. In self-similar RT/RM mixing, the type of dynamics remains the same for a>0, and has dramatically changing behaviors for $a\in \left(-2,-1\right)$. Hence, for accurate quantification of RT/RM dynamics in supernova-relevant conditions, one needs first to study the linear and nonlinear dynamics by applying acceleration patterns with a>0, and the self-similar dynamics by applying the pure RM case. One needs next to link these results (by using, e.g., artificial intelligence methods), and only then proceed to patterns with $a\in \left(-2,-1\right)$.

## 5. Summary

Supernovae are a central problem in astrophysics; they are also hypothesized as an initial value problem depicting the entire process of stellar evolution and nucleosynthesis. Rayleigh–Taylor and Richtmyer–Meshkov instabilities developing during the star’s explosion lead to intense mixing of the materials of the progenitor star and couple astrophysical to atomic scales. Based on group theory, we directly linked the conservation laws governing RT/RM dynamics to the symmetry-based momentum model and provided insight into RT/RM dynamics in a broad range of conditions.

RT/RM dynamics are driven by the specific balance of the rates of momentum gain and loss, whereas the buoyancy and drag parameters are distinct in different regimes. RT/RM dynamics are interfacial. It is single-scale in the linear regime and multi-scale in the nonlinear regime, and it has the amplitude dominance and the merge mechanisms of transition to self-similar mixing. RT/RM self-similar mixing can vary from super-ballistics to sub-diffusion, depending on the acceleration and retains memory of deterministic conditions for any acceleration. The theory outcomes explain the richness of structures observed in supernovae, contribute to the design of laboratory experiments to quantify RT/RM dynamics, and suggest that supernova evolution is a non-equilibrium process directed by the arrow of time. 

## Figures and Tables

**Figure 1 entropy-24-00829-f001:**
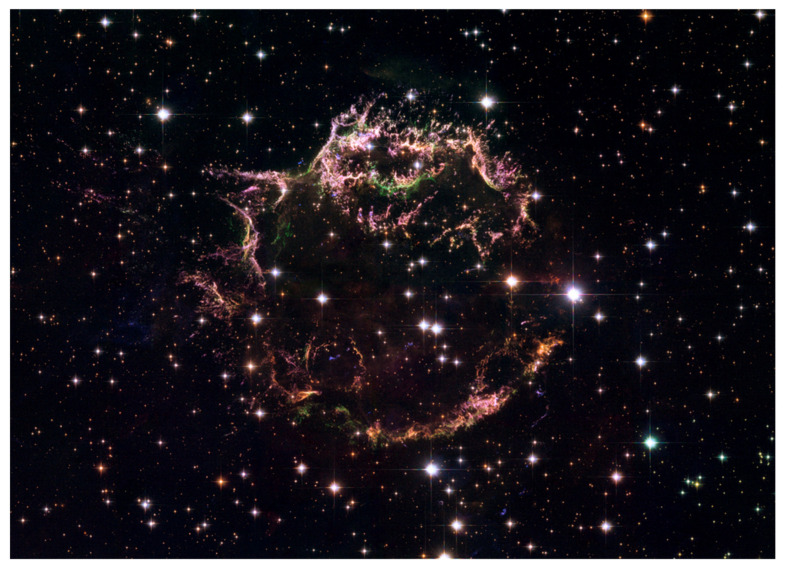
Cassiopeia A supernova remnants with filaments caused by fluid instabilities and interfacial mixing developing at the supernova blast. The colors in the filaments represent chemical compositions.

**Figure 2 entropy-24-00829-f002:**
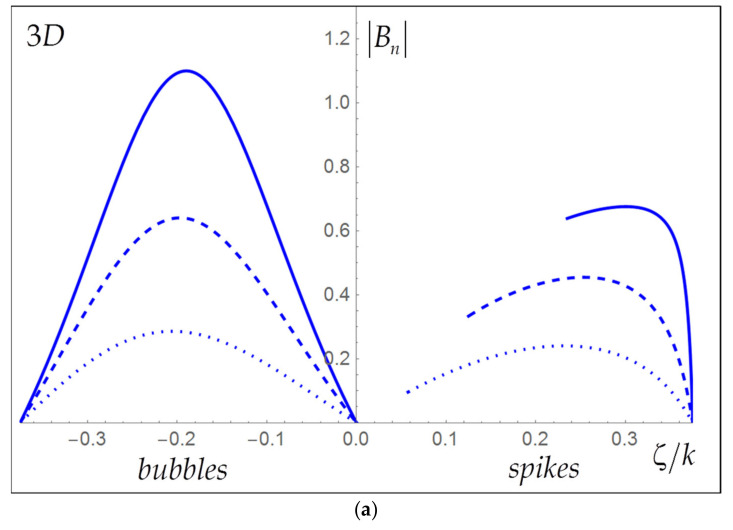
Dependence of the buoyancy parameter (**a**) and the drag parameter (**b**) on the interface morphology (curvature) for bubbles (**left**) and spikes (**right**) in the nonlinear regime for the Atwood numbers equal 0.9 (solid), 0.6 (dashed) and 0.3 (dotted) in 3D flow with hexagonal symmetry.

**Figure 3 entropy-24-00829-f003:**
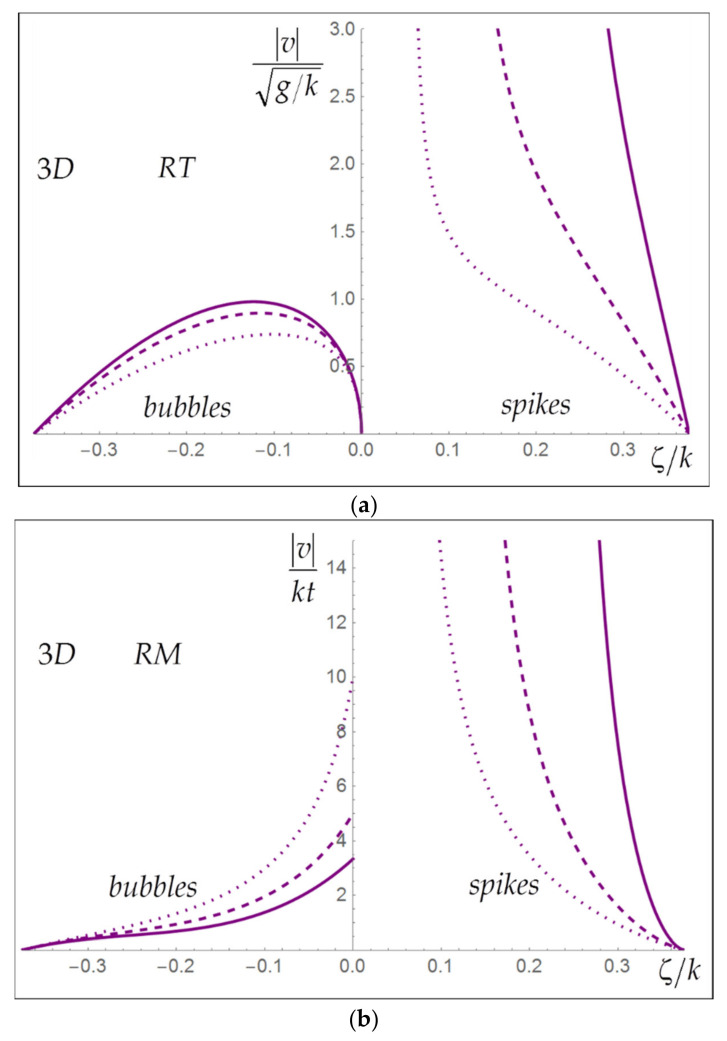
Dependence of Rayleigh–Taylor (**a**) and Richtmyer–Meshkov (**b**) solutions for bubbles (**left**) and spikes (**right**) on the interface morphology (curvature) in the nonlinear regime for the Atwood numbers equal 0.9 (solid), 0.6 (dashed) and 0.3 (dotted) in 3D flow with hexagonal symmetry.

**Table 1 entropy-24-00829-t001:** The buoyancy and the drag parameters in the linear regime.

Buoyancy	$B_{l}=-4A\xi$
Drag	$C_{l}=A/2$
Quantities	$\begin{array}{l} h\left(t\right)=Z\left(t\right),\;v\left(t\right)=\dot{Z},\;L=k^{-1},\\ \zeta \left(t\right)=Z\left(t\right)k^{2}/4,\;\xi \left(t\right)=\zeta \left(t\right)/k,\;\left|\xi \left(t\right)\right|<<1 \end{array}$
Range	$\begin{array}{l} Bubbles:\;B_{l}\geq 0,\;C_{l}\geq 0,\;\xi \left(t\right)\leq 0,\;v\left(t\right)\geq 0,\;Z\left(t\right)\geq 0.\\ Spikes:\;B_{l}\leq 0,\;C_{l}\geq 0,\;\xi \left(t\right)\geq 0,\;v\left(t\right)\leq 0,\;Z\left(t\right)\leq 0. \end{array}$

**Table 2 entropy-24-00829-t002:** The buoyancy and the drag parameters in the nonlinear regime.

Buoyancy	$B_{n}\left(A,\xi \right)=\frac{-2A\xi \left(9-64\xi^{2}\right)}{\left(3+10A\xi -128A\xi^{3}\right)}$
Drag	$C_{n}\left(A,\xi \right)=\frac{\left(9A-48\xi +64A\xi^{2}\right)}{\left(9-64\xi^{2}\right)\left(3+10A\xi -128A\xi^{3}\right)}$
Quantities	$\begin{array}{l} h\left(t\right)=Z\left(t\right),\;v\left(t\right)=\dot{Z},\;L=k^{-1},\\ \xi =\zeta /k,\;\xi \in \left[-\left|\xi_{cr}\right|,\;0\right]\cup \;\left[\xi_{A},\;\xi_{cr}\right] \end{array}$
Range	$\begin{array}{l} Bubbles:\;B_{n}\geq 0,\;C_{n}\geq 0,\;\xi \in \left[-\left|\xi_{cr}\right|,\;0\right],\;\xi \leq 0,\;v\left(t\right)\geq 0,\;Z\left(t\right)\geq 0.\\ Spikes:\;B_{n}\leq 0,\;C_{n}\geq 0,\;\xi \in \;\left[\xi_{A},\;\xi_{cr}\right],\;\xi \left(t\right)\geq 0,\;v\left(t\right)\leq 0,\;Z\left(t\right)\leq 0. \end{array}$

**Table 3 entropy-24-00829-t003:** The buoyancy and the drag parameters in the mixing regime.

Buoyancy	$B_{m}\in \left(-\infty ,0\right]\cup \left[0,+\infty \right)$
Drag	$C_{m}\in \left(-\infty ,0\right]\cup \left[0,+\infty \right)$
Quantities	$h\left(t\right)=Z\left(t\right),\;v\left(t\right)=\dot{Z},\;L=\left|h\right|$
Range	$\begin{array}{l} Bubbles:\;B_{m}\geq 0,\;C_{m}\geq 0,\;v\left(t\right)\geq 0,\;Z\left(t\right)\geq 0.\\ Spikes:\;B_{m}\leq 0,\;C_{m}\geq 0,\;v\left(t\right)\leq 0,\;Z\left(t\right)\leq 0. \end{array}$

## Data Availability

Our methods, results and data are available to interested researchers upon reasonable request.

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
