# Peer review of "Supernovae and the Arrow of Time"

_entropy, 2022, doi:10.3390/e24060829_

Round 1

Reviewer 1 Report

Ref: entropy-1589123

Corresponding author: Snezhana Abarzhi

Report:

The submitted paper concerns the application of Group theory to Rayleigh-Taylor and Richmyer-Meshkov (RT/RM) instabilities developing during stars explosions in order to describe some peculiar characteristics of structures observed in supernovae. 

The work is in the mainstream of one of authors and, on this side, provides an interesting development of RT/RM instabilities study. 

I have nothing against its publication on Entropy, of course calculations results are on authors responsibility since the very involuted matter.   

I have anyway some minor suggestions. 

- Introduction and “governing equations” paragraph (pages 1-3) show many repetitions with respect other papers of same authors (i.e. among them: S. I. Abarzhi, et al., Supernova, nuclear synthesis, fluid instabilities, and interfacial mixing  Proceedings of the National Academy of Sciences Sep 2019, 116 (37) 18184-18192;  Hill DL and Abarzhi SI (2022) On Rayleigh-Taylor and RichtmyerMeshkov Dynamics With InverseQuadratic Power-Law Acceleration. Front. Appl. Math. Stat. 7:735526).

It could have been more fluent a brief description of the problem and a description of equations that represent the basis of the study described in the paper, citation to different paper would represent the normal process in order to settle the discussion.

- Graphics of the paper is often not easy. For reader is quite difficult to follow the mathematics since formulas and comments often alternate along the text. Many equations have no spacing (see for example lines 99, 104, 194, 200 and so on). 

 - It could have been useful to insert some resuming tables at the end of each section (for example pages 6, 8, 11) since reader can get lost in order to manage all parameters that have been used along the text.

- Figure 2 would have been more satisfactory if splitted in different panels. 

- At page 7, line 235 there is a misprint, it is not “Riccarti” but “Riccati” equation.

- Page 9: there are different typefaces used along the text, probably due to some cut and paste. 

- References could be improved, for example they could be consider some recent papers like:

(2021) Richtmyer–Meshkov instability on two-dimensional multi-mode interfaces. Journal of Fluid Mechanics 928, A37

Ye Zhou et al. (2021) Rayleigh–Taylor and Richtmyer–Meshkov instabilities: A journey through scales. Physica D: Nonlinear Phenomena 423, 132838.

 Youngs et al. (2020) Early Time Modifications to the Buoyancy-Drag Model for Richtmyer–Meshkov Mixing. Journal of Fluids Engineering 142:12.

Youngs et al. (2020) Buoyancy–Drag modelling of bubble and spike distances for single-shock Richtmyer–Meshkov mixing. Physica D: Nonlinear Phenomena 410, 132517.

Shilling O., (2020) Progress on Understanding Rayleigh–Taylor Flow and Mixing Using Synergy Between Simulation, Modeling, and Experiment. Journal of Fluids Engineering 142:12.

Author Response

Dear Colleagues, Kindly find the List of Changes and the Response to the Referee. Thank you for consideration. Snezhana Abarzhi

=========================

List of Changes

  1. The Title and the Abstract are modified to improve the presentation of our work.
  2. Section 1 is modified to improve the problem formulation and respond the Referees’ comments.
  3. Section 2 is modified to improve the method presentation and respond the Referees’ comments.
  4. Section 3 is modified to improve the presentation of the results and respond the Referees’ comments.
  5. Section 4 is modified to improve the presentation of the theory outcomes.
  6. Section 5 is modified to improve the presentation of the summary of the work.
  7. Section 6 is modified to reflect the scope of our work and to improve the summary presentation.
  8. Section 7 – References: [51-58] are added.
  9. Section 8 – Tables: Table 1, Table 2, Table 3 are added to present the obtained results.
  10. Section 9 – Figures: Figure 1, Figures 2a,b, Figures 3a,b are added to illustrate our results.
  11. The detailed list of changes is the following.
  • In Title: Title is slightly modified, contact information is provided.
  • In Abstract: The Abstract is improved to better present the scope and the content of our work.
  • In Section 1: The presentation of the introduction is improved, including the problem formulation (Par 1), the astrophysical problem (Par 2), the relation to the arrow of time (Par 3, Par 4), the focus of our work (Par 5), the outline of results (Par -1).
  • In Section 2: The presentation of the method of our work is improved, including the scope (Par 1), the governing equations (Par 2), the properties of RT/RM dynamics (Par 3), and the key theoretical questions (Par -1). The presentation of equations is improved in (1.1,1.2).
  • In Section 3: The presentation of the results is improved.
  • In 3.1: The connection to traditional buoyancy-drag models is provided (Par -2). The link between the group theory implementation is emphasized (Par -1). The presentation of equations is improved in (2.1,2.2.1,2.2.2).
  • In 3.2: The presentation is overall improved. Table 1 is added (Par 1). The dependence on deterministic conditions is added (Par 2). The presentation of equations is improved in (3.1.1-3.1.3).
  • In 3.3: The presentation is overall improved. Table 1 and Figure 2a,b are added (Par 3). The dependence on deterministic conditions is added (Par 6). Details on special solutions are provided (Par 7). Figure 3a,b is added (Par 8,9). The presentation of equations is improved in (3.2.1-3.2.4).
  • In 3.4: The presentation is overall improved (Par 1,2,3).
  • In 3.5: The presentation is overall improved. Table 3 is added (Par 1). The presentation of equations is improved in (3.3.1-3.3.3).
  • In 3.5: The presentation is overall improved (Par 1,2,3).
  • In Section 4: The presentation of theory outcomes is improved.
  • In 4.1: The Sub-Section is added to discuss fundamentals of RT/RM dynamics (Par 1,2,3).
  • In 4.2: The presentation of astrophysical aspects is improved (Par 1,2).
  • In 4.3: The discussion of scaled laboratory experiments is improved (Par 1).
  • In Section 7: References are updated in [8,49]. New References are added in Refs.[51-58].
  • In Section 8: Table 1, Table 2, Table 3 are added.
  • In Section 9: Figure 1 is updated. Figure 2a,b and Figure 3a,b are added.

Reviewer 2 Report

This is a paper attempting to use  group theory for the investigation Rayleigh-Taylor and Richtmyer-Meshkov instabilities developing during the supernova explosion. The proper framework for such an investigation is the framework of general relativity due to strong gravitational field that develop during the star collapse and supernova explosion. Yet the authors not only assume a  Newtonian formulation of gravity but also assume a specific power law dependence of the gravitational acceleration $g=G t^a$. These assumptions are clearly incorrect and so are the results of this paper. I thus do not recommend publication.

Author Response

(The authors gave the same response as above.)

Reviewer 3 Report

In this article, Supernova and the arrow of time is argued to explore the process of stellar evolution and nucleosynthesis. In particular, Rayleigh-Taylor (RT) and Richtmyer-Meshkov (RM) instabilities during the the star explosion are investigated to study intense mixing of the star materials and couple astrophysical to atomic scales. The discussions could be interesting. Moreover, the mathematical analyses might be useful for the related works. Hence, if the following points are reconsidered, this paper could worthy of being published. 

1) There might exist the past related works on the relation between supernovae and the concept of the arrow of time. By comparing with these preceding studies, the new ingredients and significant progresses of this work should be stated more explicitly and in more detail. That is, the differences between this paper and the past ones should be described in more detail and more clearly. 

2) The RT/RM problem is studied based on group theory by directly linking the conservation laws governing RT/RM dynamics to symmetry-based momentum model. Particluarly, the model parameters are exactly derived in the scale-dependent and scale-invariant regimes, and the special self-similar class for RT/RM mixing with variable acceleration is analyzed. From these investigations, what physics can be deduced? 

3) It is stated that the theory outcomes are consistent with observations of supernova remnants, and that therefore they yield the design of laboratory experiment for quantification of RT/RM dynamics. Through this consequence, how can the astrophysics of supernovae be related to the nature of time including its direction (i.e., arrow)? 

4) As small points, there are several overflows of mathematical expressions in this manuscript. These should be fixed before publication. 

Author Response

(The authors gave the same response as above.)

Reviewer 4 Report

The paper Supernovae and the Arrow of Time presents its analysis of complex fluid instabilities during supernovae in terms of group theory and symmetries. To my knowledge the approach presented here is novel and has not yet appeared in any publication on SN hydrodynamics of which I am aware. I have found no errors in their calculators and I believe their conclusions to be sound.  The authors' analysis yields insights into the turbulent SN environment without resorting to numerical simulation. I believe this paper is worthy of publication.

Author Response

(The authors gave the same response as above.)

Round 2

Reviewer 1 Report

The paper in the present form can be published.

Author Response

The manuscript is modified to improve the English in the manuscript.

Reviewer 2 Report

The authors have failed to address my points in a satisfactory manner. For example they continue to use Newtonian mechanics to describe the strong gravity of supernova explosions. Thus I do not recommend publication.

Author Response

(The authors gave the same response as above.)

Reviewer 3 Report

The authors' answers to the review report are appreciated very much. 
In the revised manuscript, the points suggested in the review report 
have been reconsidered. Thus, this paper can be accepted for publication 
in Entropy. 

Author Response

(The authors gave the same response as above.)
